# Trade-offs during the COVID-19 pandemic: A discrete choice experiment about policy preferences in Portugal

**Luís Filipe**[1]*, **Sara Valente de Almeida**[2], **Eduardo Costa**[3], **Joana Gomes da Costa**[4], **Francisca Vargas Lopes**[5,6], **João Vasco Santos**[7,8,9]

1 Department of Health Research, Lancaster University, Lancaster, United Kingdom, 2 School of Public Health, Imperial College London, London, United Kingdom, 3 Nova School of Business and Economics, Universidade Nova de Lisboa, Carcavelos, Portugal, 4 School of Business and Economics, University of Porto, Porto, Portugal, 5 Department of Public Health, ErasmusMC, Rotterdam, Netherlands, 6 Erasmus Centre for Health Economics Rotterdam, Erasmus University Rotterdam, Rotterdam, Netherlands, 7 Public Health Unit, ACES Grande Porto VIII – Espinho/Gaia, Vila Nova de Gaia, Portugal, 8 Faculty of Medicine, MEDCIDS – Department of Community Medicine, Information and Health Decision Sciences, University of Porto, Porto, Portugal, 9 CINTESIS - Centre for Health Technology and Services Research, Porto, Portugal

☉ These authors contributed equally to this work.
* l.filipe@lancaster.ac.uk

**Data Availability Statement:** The dataset is anonymous and does not allow the identification of individual respondents. However, based on the informed consent signed by respondents and on

## Abstract

The need to control the sanitary situation during the COVID-19 pandemic has led governments to implement several restrictions with substantial social and economic impacts. We explored people's trade-offs in terms of their income, life restrictions, education, and poverty in the society, compared to their willingness to avoid deaths. We applied a web-based discrete choice experiment to elicit preferences of the Portuguese citizens for these attributes and computed the marginal rate of substitution in terms of avoided deaths. We recorded 2,191 responses that faced the possibility of having 250 COVID-19 related deaths per day as the worst possible outcome from the choice levels presented. Estimates suggested that individuals would be willing to sacrifice 30% instead of 10% of their income to avoid approximately 47 deaths per day during the first six months of 2021. For the same period, they would also accept 30% of the students' population to become educationally impaired, instead of 10%, to avoid approximately 25 deaths; a strict lockdown, instead of mild life restrictions, to avoid approximately 24 deaths; and 45% of the population to be in risk of poverty, instead of 25%, to avoid approximately 101 deaths. Our paper shows that avoiding deaths was strongly preferred to the remaining societal impacts; and that being a female, as well as working on site, led individuals to be more averse to such health hazards. Furthermore, we show how a DCE can be used to assess the societal support to decision-making during times of crisis.

## 1. Introduction

The management of the COVID-19 pandemic was characterised by difficult choices. In several phases of the pandemic, most governments were compelled to implement strict lockdowns.

the authorization granted by NOVA School of Business and Economics Ethics Committee, data cannot be shared in a public repository to respect data protection regulation. A minimal data set for replication of the study findings is available upon request to the NOVA School of Business and Economics Ethics Committee (research.office@novasbe.pt) and the corresponding author.

**Funding:** This report is independent research funded by the National Institute for Health Research (NIHR) Applied Research Collaboration North West Coast (ARC NWC). The views expressed in this publication are those of the authors and not necessarily those of the NHS, NIHR, or the Department of Health and Social Care. There are neither financial nor non-financial conflicts. No ethical considerations apply.

**Competing interests:** The authors have declared that no competing interests exist.

These lockdowns made people stay at home, often only being allowed to leave for essential reasons. It also implied the temporary closure of schools, retail, and many other activities. All these measures had a clear and negative impact on families' income, education, and social lives [1,2]. These were significant policy choices that implied difficult trade-offs. It is not clear whether, when making these decisions, governments were able to understand and reflect society preferences and follow the scientific evidence, at the same time.

Understanding individuals' preferences in this context is extremely important as the efficacy of contingency measures depended heavily on individuals' behaviours. Literature has shown that it is challenging to capture these preferences [3,4]. To overcome this, researchers have been increasingly using Discrete Choice Experiments (DCE) to understand preferences and inform policy decisions, which is especially relevant in healthcare studies [5].

DCEs allow the study of choices with a high degree of flexibility. This tool is implemented by creating hypothetical games where individuals choose between options, implicitly revealing their preferences. Such knowledge of preferences, inherently linked to beliefs and behaviours, is a key piece of information to guide government decisions. It is particularly relevant in the context of a crisis, when unprecedented measures must be adopted in short time frames [6,7]. DCE studies have been used in the context of COVID-19, not only to measure how the citizens value the costs and benefits of lockdown policies [8–12] but also to understand potential barriers and preferences towards the uptake of COVID-19 measures, such as testing, contact-tracing and vaccination [13–16].

The objective of this study was to use a DCE to assess individuals' preferences regarding the consequences of COVID-19 related policies in Portugal and to explore which of their characteristics might be associated with heterogeneous preferences among the population.

We evaluated preferences in terms of five attributes: number of daily deaths, loss of household income, life restrictions (limitations to individual freedom), educational impairment, and levels of poverty in the society. Our study provides new information on how Portuguese citizens considered these policy trade-offs, by estimating their willingness to avoid deaths relative to the other attributes. We examined heterogeneous preferences based on an extensive set of individual characteristics collected in our survey through i) an unconditional subgroup analysis, and ii) a latent class model. Furthermore, we were able to capture preferences at a particularly challenging moment of the pandemic management. Our survey was applied during the third and most deadly COVID-19 wave in Portugal. The period that led to the third wave was marked by stronger public debate around restrictions. While conducting the survey, COVID-19 numbers were growing fast, and the Portuguese National Health Service (NHS) displayed signals of being overwhelmed. Thus, the survey captured preferences of individuals at a crucial phase of decision-making by the government.

## 2. The COVID-19 pandemic in Portugal

On March 18[th], 2020, following the rapid increase of COVID-19 cases in European countries, Portugal declared the beginning of the "State of Emergency", allowing a set of exceptional legal provisions to be implemented in fighting the pandemic. Among other measures, over a six-week period, only citizens from the essential services could work outside their homes. All other citizens including children had to be confined and were only allowed to leave their homes in exceptional circumstances.

In mid-April, at the worst stage of the first wave, Portugal reached 1,516 daily new cases of COVID-19 and 37 daily deaths (Portugal has approximately 10 million inhabitants). In May, the numbers started to decrease significantly. Most measures adopted in the first stage of the pandemic were gradually scaled down in the following months. The summer was quiet, with

the number of new daily cases stabilising around 300 and the number of daily deaths in values below five.

The number of daily cases started to slowly rise again in September 2020, with the growth hitting an exponential trend in October. On November 9[th], the country went back to a "State of Emergency". In this second wave, the maximum number of new daily cases increased to 6,994 (November 19[th]) and the highest number of daily deaths was 98 (December 13[th]).

After Christmas and New Year's Eve celebrations, when restrictions were partially relaxed and gatherings were allowed, the country saw a sudden rise in the number of cases that would lead to the third and worst wave of the pandemic. At this time, Portugal was among the first western countries to be hit with the third wave and recorded the highest levels of cases and deaths per capita at a global scale. On January 15[th], the government reinstated a nationwide lockdown due to the surge in new daily cases. The highest daily number of new cases and deaths occurred on January 28[th] with 16,432 new cases and 303 deaths. Portugal remained in lockdown for the next few months. The "State of Emergency" was lifted on May 1[st], 2021. From then onwards, the restrictive measures were gradually dropped.

Our survey was distributed between January 12[th] and March 14[th], 2021, capturing the most precarious period of the pandemic in Portugal. During the survey collection period, vaccination was at a very early stage—in the peak of the third wave, only about 2.5% of the population had received a first dose of the vaccine.

## 3. Methodology

### 3.1. DCE

We used a DCE to evaluate the preferences of the Portuguese citizens regarding the possible effects of different COVID-19 policies. This is a recognized methodology to elicit latent preferences and has been used when presenting the respondents with difficult moral trade-offs [17,18].

### 3.2. Attributes and levels

The study used five attributes: number of COVID-19 related deaths, loss of household income, educational impairment, life restrictions, and poverty levels in the society (Table 1). Each attribute was divided into three levels of low, medium, and high intensity. Based on international literature, other DCEs were used to inform this study [8,12]. Their attributes were adapted to the Portuguese context based on the findings of a previous COVID-19 survey [19] and authors' perception of policy relevance of the outcomes. For each attribute we defined the high (worst) level based on a critical evaluation of the data and the policies implemented at the time of the survey [20]. For the quantitative attributes, medium and low levels were constructed by

**Table 1. Attributes and levels of the survey for the period between January and June 2021.**

| Attributes | Units | Low | Medium | Maximum |
|---|---|---|---|---|
| COVID-19 related deaths | Daily deaths attributed to the pandemic either directly (i.e., COVID-19 infection) or indirectly (i.e., limited response by the health care system) | 50 | 150 | 250 |
| Household income lost | % decrease in respondent's household income motivated by the pandemic | 10 | 20 | 30 |
| Educational impairment | % of students permanently affected by the pandemic in the school population | 10 | 20 | 30 |
| Life restrictions | Type of restrictions in place for all the citizens | Large-events restrictions and no bars/ clubs | Night curfew and max 5 people gatherings | Full lockdown |
| Poverty level | % of the population in risk of poverty | 25 | 35 | 45 |

adopting linear decrements of a third to a fourth of the highest level. In terms of the duration of the effects, we defined they would last for a period of six months (January to June 2021).

In Portugal, the number of COVID-19 deaths was in the spotlight every day since the beginning of the pandemic. This led us to include COVID-19 related deaths as the baseline attribute in our survey, i.e., the marginal rate of substitution (MRS) was defined as the number of deaths to be traded for other attributes. The highest level for this attribute was defined as Portugal's linear equivalent to the highest daily number of deaths observed in Spain in 2020, during that country's severe first wave.

With the consecutive lockdowns, the economy slowed down considerably (GDP in 2020 decreased by 8.4%). This led to a considerable loss of household income for many people, either because they owned a business that was affected or because they were laid off [21–23]. The second attribute was the percentage decrease in the respondent's household income related with the pandemic. The levels for this attribute were 10%, 20% and 30%. The choice of maximum loss of 30% considered that laid off employees or companies that had to shut down would still have access to 70% of their income, through social security schemes.

Schools belong to the wide range of institutions that closed during the first times of the pandemic. Portuguese high schools were closed for 97 days, between January 2020 and May 2021 [2]. As a third attribute, we included the percentage of the school population that would become permanently affected by the pandemic. To some extent, this percentage reflects the inequalities associated with schools' closure, distance learning and other mitigation strategies. The levels for this attribute were 10%, 20% and 30%. The maximum level was chosen in accordance with a report from the World Bank [24] estimating the number of students who may suffer from losses because of the school closures.

Strict restriction policies impacted the regular day-to-day life of the population. Being deprived of social interactions can be burdensome [25]. The fourth attribute was the level of life restrictions imposed by the government. The levels of the attributes were chosen to reflect different scenarios that occurred during the first year of the pandemic, as described in section 2. The low intensity level contemplated only large events' restrictions and bars and discos closed at night; the medium level was characterised by a mandatory curfew from 23h00 do 5h00 and gatherings restricted to five persons; the high level was a full lockdown, with people working from home, circulation restrictions and schools shut down.

The economic collapse driven by life restriction measures has led to an inevitable loss of global wealth. The potential measures to fight the pandemic ended up pushing people into poverty. The fifth attribute was the percentage of the population in risk of poverty. The levels for this attribute are 25%, 35% and 45%. The maximum level corresponded to the approximate percentage of the population at risk of poverty before any social transfer, observed in the past 10 years [26]. This would correspond to a catastrophic scenario where social transfers would be unavailable.

## 3.3. Survey

The project was approved by the NOVA School of Business and Economics Ethics Committee on the 28[th] of December 2020. When opting to answer the survey, the respondents have consented to the publication of potential findings.

The data was collected through an online survey between January 12[th] and March 14[th], 2021. Most observations were collected during January. Any adult living in Portugal in this period was eligible to answer. The survey was distributed using social media and mailing lists of the University of Porto and Nova School of Business and Economics (in Lisbon). The survey was created in Qualtrics and divided in two parts. In the first part, respondents were asked

**Table 2. Example of a choice set.**

| For the period between January and June 2021 | Option A | Option B |
|---|---|---|
| Daily deaths attributed to COVID-19, directly or indirectly (e.g., delay on healthcare provision) | 150 | 50 |
| % decrease on household income motivated by the pandemic | 20 | 10 |
| % of students permanently affected by the pandemic | 10 | 20 |
| Level of restrictions to day-to-day life* | High | Low |
| % of the population in risk of poverty (living with < €6/day) | 25 | 35 |

*Low: Large-events restrictions and no bars/clubs; Medium: Night curfew and maximum 5 people gatherings; High: Full lockdown.

about their socio-demographic characteristics. The second part implemented a DCE with 16 choice sets, separated in two blocks. Participants were randomly allocated to respond to one of the two blocks of 8 choice sets. The alternatives for each set were generated by a D-Optimal design [27,28]. We conducted a pilot study between January 2nd and January 6th with 73 respondents. The pilot was used to pre-test our survey and adjust the D-Optimal design parameters. Since no changes were made to the survey, the pilot is also included in the analysis.

Table 2 shows a choice set translated to English (an original example and instructions in Portuguese can be found in S1 and S2 Figs in S1 Appendix). Before starting the DCE, respondents were informed about the scope of the survey. It was clearly stated that alternatives were hypothetical, and that respondents should disregard the potential relation between attributes. They were instructed to consider that effects would last for six months, between January and June of 2021. Respondents were also given a brief instruction on the measures included in each life restriction attribute level and on how to interpret the probability values in the other three attributes. In the first choice set respondents were given the alternative to quit.

## 3.4. Population in the survey

The survey had a total of 2,191 respondents. Key descriptive statistics are displayed in Table 3. Respondents' age ranged from18 years old up to over and 63% of our sample was represented by individuals below 35 years old. Most of the respondents were women (approximately 63%) and had a bachelor's degree (approximately 32%), or a master's degree or higher (approximately 40%). Individuals in the sample displayed a median monthly equivalised income of around €1,250. The poorer 25% (excluding missing answers) received an equivalent household income of less than €750 per individual; the 25%-50% equivalent household income quartile corresponded to €751-€1,250 per individual; the 50%-75% quartile was between €1,251-€1,750 per individual; and the 75%-100% quartile was above €1,750.

Nearly 42% of the individuals in the sample were living in Porto in comparison with approximately 23% of the respondents living in Lisbon and the remaining elsewhere in the country. Households presented a mean of three individuals, and approximately 60% of all households had no children. Our sample was mainly constituted by students (approximately 28%) and public servants (approximately 21%). Half of the sample (approximately 51%) was working from home at the time of the survey.

## 3.5. Analysis

We performed a fourfold analysis. First, we ran a conditional logit model [29], with all attribute levels coded as binary. Second, we re-estimated the model with number of daily deaths coded as

**Table 3. Key descriptive statistics for the sample and comparison with the Portuguese population.**

| | SAMPLE | PORTUGAL |
|---|---|---|
| | N (%) | N (%) |
| **Age** | | |
| 19–25 years | 646 (29.48) | 557,119 (5.41) |
| 26–35 years | 739 (33.73) | 1,114,060 (10.82) |
| 36–45 years | 426 (19.44) | 1,435,773 (13.94) |
| 46–55 years | 219 (10.00) | 803,727 (7.81) |
| 56–65 years | 135 (6.16) | 1,423,032 (13.82) |
| 66–75 years | 21 (0.96) | 1,178,309 (11.44) |
| 76–85 years | 4 (0.18) | 788,663 (7.66) |
| > 85 years | 1 (0.05) | 328,066 (3.19) |
| No answer | - | - |
| **Gender** | | |
| Female | 1,379 (62.94) | 5,430,098 (52.48) |
| Male | 807 (36.83) | 4,917,794 (47.52) |
| No answer | 5 (0.23) | - |
| **Education attainment level** | | |
| No college Degree[1] | 579 (26.43) | -78.8 |
| Bachelor's degree[2] | 711 (32.45) | (21.2)[3] |
| Master's Degree or Higher[4] | 894 (40.80) | |
| No answer | 7 (0.32) | |
| **Monthly equivalent household income** | | |
| < €750 | 497 (22.68) | N/A[5] |
| €751-€1,250 | 487 (22.23) | |
| €1,251-€1,750 | 515 (23.51) | |
| > €1,751 | 518 (23.64) | |
| No answer | 174 (7.94) | |
| **Area of residence** | | |
| Porto | 917 (41.85) | 1,729,390 (16.80) |
| Lisbon | 512 (23.37) | 2,270,980 (22.05) |
| Other | 762 (34.78) | 6,296,711 (61.15) |
| **Number of persons in household** | | |
| Mean | 3.01 | 2,5 |
| **Number of children in household** | | |
| Younger than 6 | 285 (13.01) | N/A[5] |
| Both older and younger than 6 | 101 (4.61) | |
| Older than 6 | 446 (20.36) | |
| None | 1,287 (58.74) | |
| No answer | 72 (3.29) | |
| **Occupation** | | |
| Unemployed | 108 (4.93) | N/A[5] |
| Student | 620 (28.30) | |
| Student-Worker | 21 (0.96) | |
| Researcher | 34 (1.55) | |
| Public servant | 457 (20.86) | |
| Retired | 33 (1.51) | |
| Large enterprise employee | 326 (14.88) | |
| SME employee | 375 (17.12) | |
| Self-employed | 194 (8.85) | |
| Other | 9 (0.41) | |
| No answer | 14 (0.64) | |

*(Continued)*

**Table 3.** (Continued)

| | SAMPLE | PORTUGAL |
|---|---|---|
| | N (%) | N (%) |
| **Home Office** | | |
| No | 1,070 (48.84) | N/A[5] |
| Yes | 1,121 (51.16) | |
| No answer | - | |

[1] - Levels 0, 1, 2, and 3 (ISCED).

[2] - Level 6 (ISCED).

[3] - Information on University Education was provided as an aggregate measure.

[4] - Level 7 and 8 (ISCED).

[5] - Non-Available Information.

a continuous variable, to compute the MRS using deaths as a numeraire. Third, we performed unconditional subgroup analysis, through the estimation of a conditional logit for the relevant characteristics of the population collected in the survey. Fourth, we conducted a latent class analysis [30,31] to account for class preferences' heterogeneity. We tested models from 2 up to 10 latent classes. Three classes were selected to be presented in the manuscript based on the Consistent Akaike information criterion (CAIC) and the Bayesian information criterion (BIC).

Data analysis and the creation of the D-optimal design were made with STATA 17 software.

## 4. Results

### 4.1. Main results

Fig 1 shows the results of the conditional logit model and respective MRS. As expected, coefficients for high and medium levels of the attributes are negative and statistically significantly different from the low level, which serves as the baseline level. However, despite the difference relative to the low level, coefficients for medium and high levels for the education and life restrictions attributes have similar magnitudes.

From the relative size of coefficients for different attributes we can conclude that 250 daily deaths due to COVID-19 was the attribute level that affected preferences the most (relative to 50 deaths due to COVID-19). Considering only the maximum levels of each attribute, next comes 45% of the population at the risk of poverty (relative to 25%), followed by 30% loss of respondent's household income (relative to 10%), education impairment in 30% of the school population (relative to 10%), and the highest level of life restrictions–a full lockdown (relative to mild life restrictions).

Based on the analysis of the MRS, the respondents in this survey were willing to sacrifice 30% of their income (instead of 10%) for six months to avoid approximately 47 daily deaths during the same period. They would also accept 30% of the school population to become educationally impaired (instead of 10%) to avoid approximately 25 daily deaths; going from a situation in which there were only restrictions to large-events, bars, and clubs to a full lockdown to avoid approximately 24 daily deaths; and 45% of the population to be at the risk of poverty (instead of 25%) to avoid roughly 101 daily deaths, all for six months.

### 4.2. Subgroup analysis

Fig 2 shows the MRS for a range of subgroups. While estimates of MRS (coefficients available in S3 Table in S1 Appendix) differed in magnitude for several of the characteristic examined,

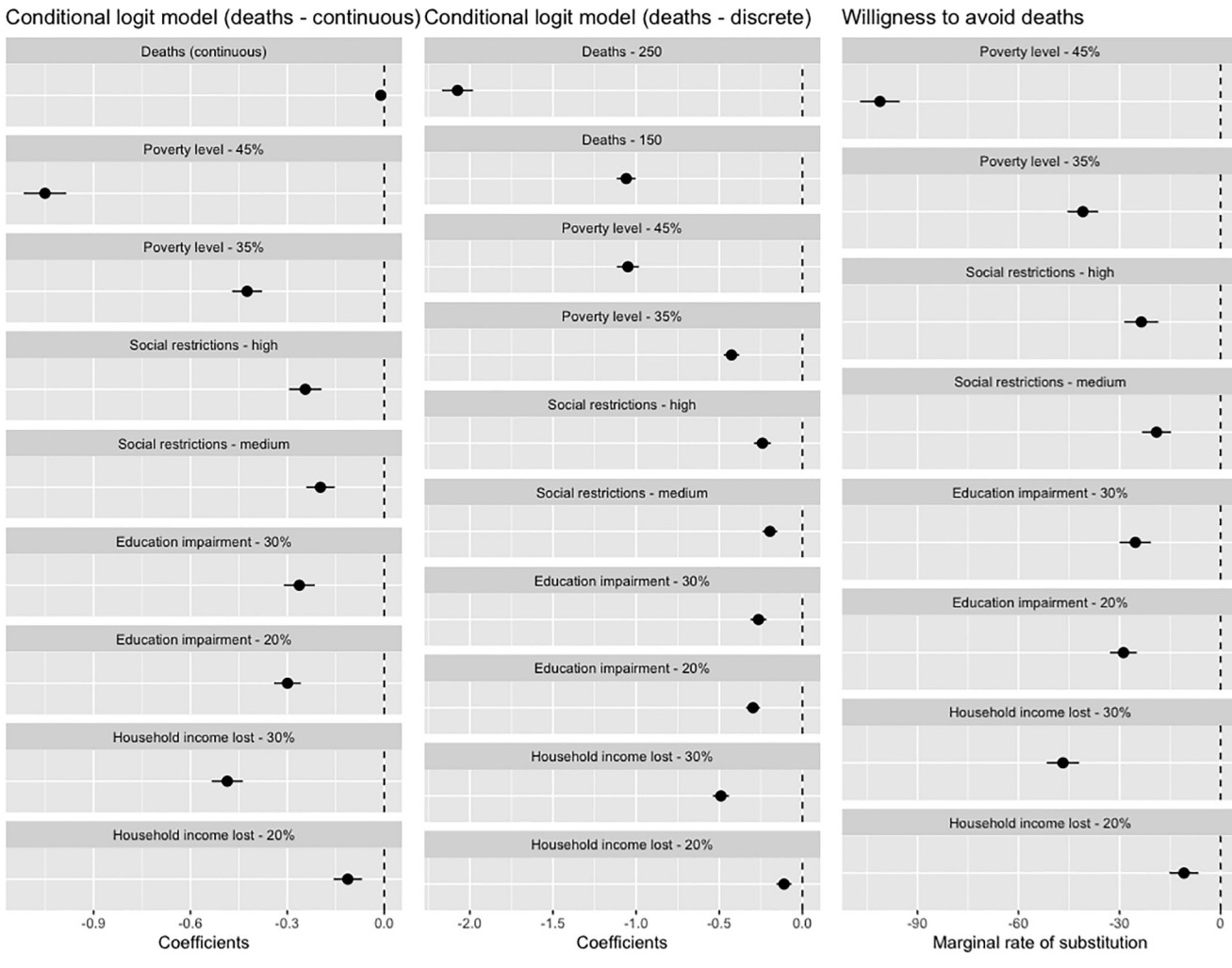

**Fig 1. Coefficients from the main conditional logit model and the respective MRS.** Note: S2 Table in S1 Appendix, displays the numbers supporting these graphs.

we were reluctant to interpret those without a statistically significant difference. There were, however, two characteristics for which we find statistically significant differences in preferences: gender and working location during the pandemic.

In Fig 2, we see that women had lower MRS than men for most attributes, significantly different in terms of life restrictions and higher level of educational impairment. For example, women were willing to accept a stricter lockdown, instead of milder restrictions, to avoid approximately eight daily deaths, whereas men were willing to accept a stricter lockdown, instead of milder restrictions, to avoid approximately 53 daily deaths.

The same pattern was observed in the workplace subgroup (first graph of the second column), where people working on site have statistically significantly lower MRS for the maximum levels of education impairment and life restrictions than those working from home. For example, people working on site were willing to accept a stricter lockdown, instead of milder restrictions, to avoid approximately 15 daily deaths, whereas people working from home were willing to accept a stricter lockdown, instead of milder restrictions, to avoid approximately 32 daily deaths.

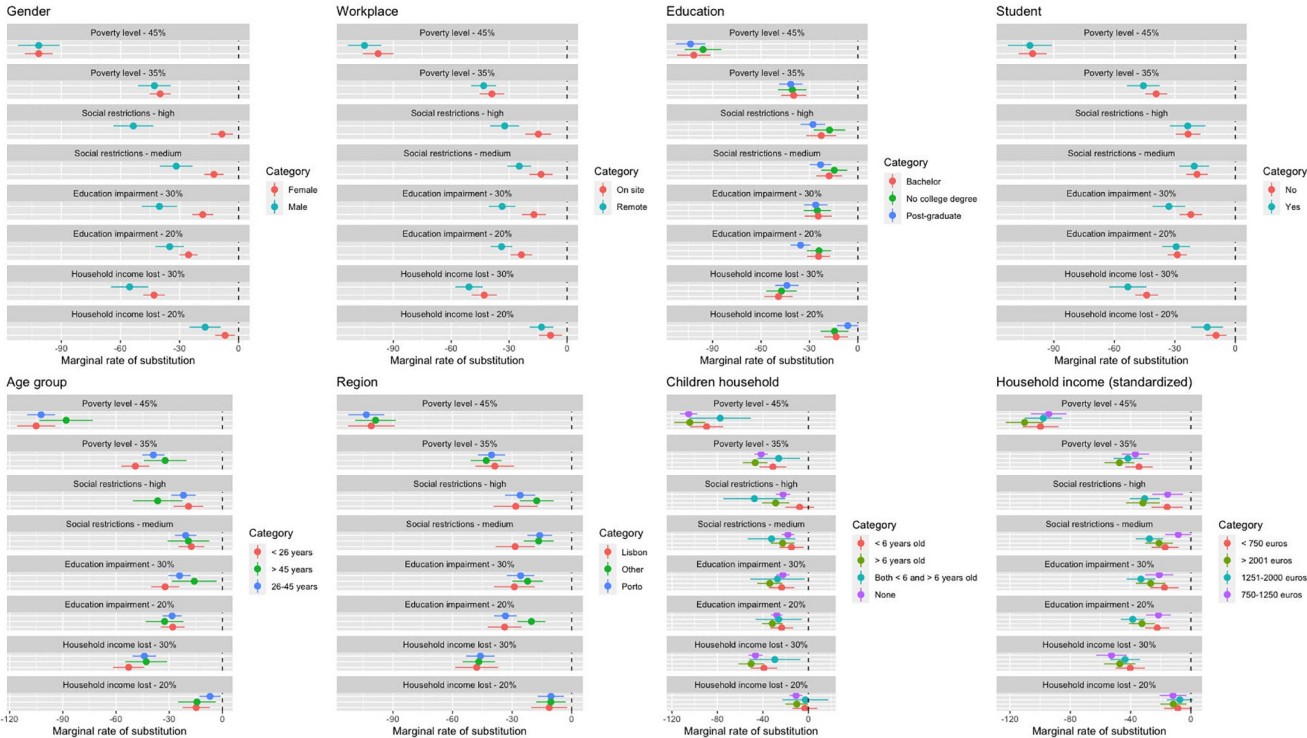

**Fig 2. Marginal rate of substitution (MRS) by subgroups.**

## 4.3. Latent class model

Three latent classes of patient preferences were identified. Results are shown in Table 4. People with Class 1 preferences, who make 28% of the sample, gave the lowest importance to COVID-19 deaths. According to Fig 3, they gave higher relative value to education, life restrictions, and risk of poverty than the other two classes. People with Class 1 preferences were more likely to be male, work remotely, and to be part of a household with only children older than six years old.

People with Class 2 preferences, representing 41% of the sample, were characterized by a higher valuation of COVID-19 deaths, when compared to people with Class 1 preferences. They also had a lower valuation for household income losses, when compared with people from Class 1 and Class 3. Fig 3 shows that they gave a relatively lower importance to education, life restrictions and poverty, then people with Class 1 preferences. People with Class 2 preferences were less likely to be aged 45 years or more and to be from Porto.

Individuals with Class 3 preferences, representing 31% of the sample, had the strongest valuation toward all levels of the deaths' attribute. However, they displayed rationally inconsistent results for the education impairment and life restriction attributes. Fig 3 shows that people in this class gave the most relative importance to deaths. People from Class 3 were more likely to be older than 45 years than people with Class 1 or 2 preferences.

## 5. Discussion

We found a high number of deaths due to COVID-19 to be the most impactful attribute on individual's preferences. Respondents were willing to trade-off considerable impacts on their lives in terms of income loss and limitations to individual freedom, as well as impacts for the

**Table 4. Coefficients from the latent class model with 3 classes.**

| Attribute Levels | Class1 Coeff | 95% CI | | Class2 Coeff | 95% CI | | Class3 Coeff | 95% CI | |
|---|---|---|---|---|---|---|---|---|---|
| Deaths—150 | **-0.47** | -0.62 | -0.32 | **-1.31** | -1.52 | -1.11 | **-4.15** | -4.94 | -3.35 |
| Deaths—250 | **-0.55** | -0.81 | -0.29 | **-2.64** | -3.04 | -2.23 | **-12.47** | -15.09 | -9.86 |
| Household income lost—20% | **-0.63** | -0.84 | -0.42 | **-0.16** | -0.28 | -0.04 | 0.60 | -0.62 | 1.83 |
| Household income lost—30% | **-1.28** | -1.55 | -1.01 | **-0.60** | -0.73 | -0.47 | **-3.10** | -4.21 | -1.99 |
| Compromised education—20% | **-0.42** | -0.60 | -0.25 | **-0.41** | -0.53 | -0.28 | **-1.13** | -1.89 | -0.37 |
| Compromised education—30% | **-0.90** | -1.13 | -0.68 | **-0.54** | -0.67 | -0.41 | **3.91** | 2.71 | 5.10 |
| Life restrictions—medium | **-0.32** | -0.49 | -0.14 | 0.00 | -0.11 | 0.10 | **0.55** | 0.11 | 0.99 |
| Life restrictions—high | -0.07 | -0.23 | 0.09 | **-0.18** | -0.31 | -0.06 | 0.60 | -0.14 | 1.33 |
| Risk of poverty—35% | **-1.21** | -1.49 | -0.93 | **-0.56** | -0.68 | -0.45 | **4.36** | 3.03 | 5.69 |
| Risk of poverty—45% | **-1.60** | -1.85 | -1.36 | **-1.51** | -1.68 | -1.34 | **-4.02** | -5.38 | -2.67 |
| Class membership model | | | | | | | | | |
| Gender: Female (reference) | | | | | | | Reference Class | | |
| Gender: Male | **0.49** | 0.20 | 0.78 | 0.04 | -0.25 | 0.33 | | | |
| Age: 18–25 (reference) | | | | | | | | | |
| Age: 26–45 | -0.24 | -0.72 | 0.25 | -0.40 | -0.89 | 0.09 | | | |
| Age: >45 | **-0.58** | -1.14 | -0.03 | **-1.01** | -1.59 | -0.43 | | | |
| Working type: On site (reference) | | | | | | | | | |
| Working type: Remote | **0.31** | 0.03 | 0.59 | -0.09 | -0.36 | 0.19 | | | |
| Region: Other (reference) | | | | | | | | | |
| Region: Porto | -0.13 | -0.44 | 0.18 | **-0.33** | -0.64 | -0.02 | | | |
| Region: Lisboa | 0.05 | -0.31 | 0.42 | -0.05 | -0.42 | 0.33 | | | |
| Children in HH: <6 | | | | | | | | | |
| Children in HH: <6 & >6 | 0.09 | -0.60 | 0.77 | -0.03 | -0.73 | 0.66 | | | |
| Children in HH: Only >6 | **0.63** | 0.13 | 1.13 | 0.40 | -0.11 | 0.90 | | | |
| Children in HH: None | 0.38 | -0.05 | 0.81 | 0.31 | -0.11 | 0.74 | | | |
| Education: No college degree (reference) | | | | | | | | | |
| Education: Bachelors | 0.19 | -0.19 | 0.57 | -0.17 | -0.54 | 0.19 | | | |
| Education: Masters+ | 0.04 | -0.37 | 0.46 | -0.36 | -0.77 | 0.05 | | | |
| HH eq. income: < €750 | | | | | | | | | |
| HH eq. income: €751 – €1,250 | -0.10 | -0.47 | 0.27 | 0.00 | -0.38 | 0.37 | | | |
| HH eq. income: €1,251-€2,000 | 0.03 | -0.34 | 0.41 | 0.25 | -0.14 | 0.64 | | | |
| HH eq. income: >€2,001 | 0.13 | -0.26 | 0.52 | 0.34 | -0.07 | 0.75 | | | |
| Student: No (reference) | | | | | | | | | |
| Student: Yes | -0.09 | -0.57 | 0.39 | -0.09 | -0.58 | 0.39 | | | |
| Constant | -0.61 | -1.42 | 0.20 | 0.69 | 0.03 | 1.34 | | | |
| Class share | | 0.28 | | | 0.41 | | | 0.31 | |

Numbers in bold stand for statistically significant coefficients at 95% confidence level.

Number of observations: 33,808; Log likelihood = -8242.7859; AIC: 16609.61 CAIC: 17024.52; BIC: 16962.52.

society in terms of educational impairment and poverty, to avoid going from 50 to 250 daily deaths related to COVID-19. Levels of poverty were the second attribute in priority for our respondents, with individuals' household income losses coming in third. While the individual household income loss reported a more selfish view of the economic impacts of the pandemic, the levels of poverty captured a more altruistic perspective. The high importance given to poverty levels suggests that the respondents were very concerned with the pandemic effects on the

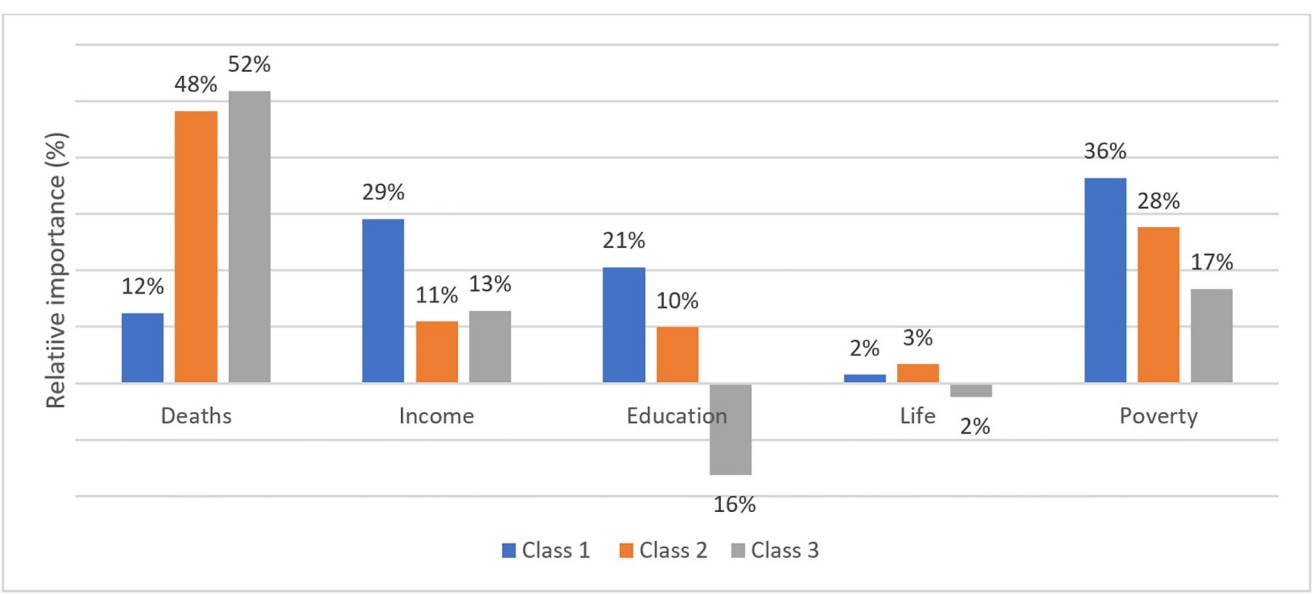

**Fig 3. Relative importance of attributes per class.** Note: The relative importance is computed by taking the absolute value of the maximum level of each attribute and dividing them by the sum of the absolute value of all the maximum levels of each attribute, per class [32]. The downward bars for "Education" and "Life" attributes in Class 3 are purely illustrative, to convey that the valuation given to those attributes' levels are contrary to rationally consistent preferences.

economical capacity of their countrymen. Subgroup results suggest that women presented lower MRS than men for the education and life restriction attributes. S4 Table in S1 Appendix, shows that these results are driven by both a stronger negative effect of deaths and a weaker negative effect of the two attributes on women's preferences, when compared to men. Several studies conducted have found general evidence that women reported to agree more and be more compliant with the restrictions imposed in different countries [33–35]. Women reported greater fear, more negative expectations about health-related consequences and were more likely to consider COVID-19 infection a serious health problem. Moreover, during the pandemic, women reported higher levels of stress and concern, as well as higher levels of overwork and informal care, compared to their male counterparts [36–38]. Our results also support this evidence since being more worried naturally increases one's MRS to stop the spreading of the virus and avoid deaths. The gender differences found in our results might also be reflecting the sample selection bias induced in our data. There is, for example, a larger proportion of women than men working on site in our sample (67.38% vs. 32.62%), as shown in S1 Table in S1 Appendix. Our findings also suggest that the MRS of those working remotely was twice as high as the ones working on site, for the maximum level of the life restrictions' attribute. These results are compatible with potential saturation felt by individuals in remote working; but they might also result from lower exposition to risk. At the same time, this can also be explained by the largest share of female respondents working on-site. Further investigation would be needed to disentangle these two potential effects.

The latent class analysis supported, to some extent, the results found in the unconditional sub-group analysis. People with Class 1 preferences were more likely to be males and working from home. While the differences were now in different attributes, it remains that men and people working from home tended to have different preferences from women and people working on site. Also, the latent class model identified a class where people were more likely to give a major importance to the deaths' attribute, to a point where only the higher levels of the

household income loss and population in risk of poverty had rationally consistent coefficients. As this is not observed in other studies, such as in Chorus (2020) [8], we believe it may be a consequence of our analysis encompassing a period of higher emergency. The class in question had a larger share of people over 45 years old which may indicate that older people developed a bigger sense of urgency than younger people, on average.

Our results are broadly in line with other DCE studies focusing on a similar question in the Netherlands [8], United States of America [12], Australia [10] and Germany [11], even though these studies collected responses at an earlier phase of the pandemic: Chorus (2020) [8] in April 2020, Reed (2020) [12] in May 2020, Manipis (2020) [10] between July and August 2020, Li (2021) [9] end of August 2020 and Mühlbacher (2022) [11] from October to November 2020). Although there are some differences between the domains studied, attributes related to COVID-19 excess mortality were consistently those with a larger effect on individual's preferences. An exception was the German sample studied by Mühlbacher (2022) [11], in which the attribute related to a decrease of individual income was the most important. Most studies also pointed to considerable heterogeneity depending on individuals' characteristics. Chorus et al. (2020) [8] found older and more highly educated Dutch citizens to be less willing to sacrifice domains as mental health, educational disadvantage, and income loss to avoid fatalities; while Li et al. (2021) [9] identified differences by age, unemployment status and ability to work from home. In line with our results, the authors found respondents who can work from home to be less willing to sacrifice some attributes, namely, to have lower willingness to stay at home to reduce unemployment and number of COVID-19 daily cases.

An important limitation of this study is the survey's lack of representativeness, which resulted from the fact that it was collected online with two universities as key channels of dissemination. Therefore, the survey ended up reaching a larger share of individuals with higher education than that of the general population. This bias may also imply a higher proportion of high-income individuals in the sample.

Preferences displayed by individuals are likely to be affected by the timing of the study. This survey was distributed during the worst wave of the pandemic in Portugal, at a time where cases and deaths were growing fast, and the Portuguese NHS was displaying strong signals of being disrupted. While having information about preferences in the peak of the pandemic gives us important insight about decision-making under extreme situations of crisis, our results may be capturing some sense of urgency, potentially not appropriate for predicting preferences in other periods.

Moreover, the COVID-19 pandemic was characterised by substantial uncertainty. The DCE setting implies using fixed levels for each attribute, throughout the experiment. These levels are defined based on the literature and on existing expectations. Some of these levels may become less optimal at a given moment, considering ongoing changes in the pandemic situation. For instance, the highest level for deaths considered in our DCE, was below the maximum level observed afterwards. Combined with the required time to design and implement, DCEs do not present substantial flexibility in volatile contexts. Additionally, during the survey collection period, the vaccination program was already being rolled out. Even though the vaccination program could potentially affect one's preferences, we do not expect this to have substantially impacted the results. In fact, in the peak of the third wave, only 2.5% of the population had received a first dose.

In the specific case of the third COVID-19 wave in Portugal our results suggest our sample supported the decisions adopted by the Portuguese government, which prioritised lives over the economy and lifestyle, by implementing a second full lockdown in the country. This is an interesting finding, particularly considering that just before the onset of the third wave there were some signs of behavioural fatigue in the Portuguese population. In this context our

findings might suggest that individuals adapt their preferences quickly when facing serious health hazards. Depending on how many deaths we assume that a lockdown could avoid, we may use our MRS results for some back-of-the-envelope calculations on the quantitative level of support of our respondents. For example, according to Flaxman et al. (2020) [39] the lockdown during the first wave of the pandemic saved around 3.1 million lives, including 470,000 in the UK, 690,000 in France and 630,000 in Italy. Even though they do not contemplate Portugal, rescaling those numbers would yield approximately 1,000 daily deaths avoided in Portugal for the same period, which is much higher than the combined MRS for all the maximum levels of the other four attributes.

Last, and while our study is a product of a convenience survey with limited generalizability, it is a first step in gathering information about the public preferences for pandemic outcomes and support for government decision-making at a time of crisis. Additionally, no feedback regarding the completion of the DCE was collected from respondents. Overall, and despite these limitations, we show that DCE are instruments available to help guide policy decision making and we defend that they are a flexible and efficient tool to systematically assess implicit preferences of the population.

## 6. Conclusion

Our results show that, during the peak of the pandemic in Portugal, individuals in this sample were willing to sacrifice substantial amounts of their income, and everyday life freedom, as well as educational outcomes and levels of poverty in the society, to avoid the daily toll of COVID-19 deaths happening at that time. Additionally, findings from the heterogeneity analysis suggest that preferences varied with characteristics such as gender and place of work. These results provide some insights to inform future research on the topic and confirm that DCEs are a useful and flexible tool to incorporate public preferences and behaviours on the design of new policies, particularly, in the context of a crisis.

## Supporting information

**S1 Appendix.**
(PDF)

## Acknowledgments

We are very grateful to Samare Huls (Erasmus University Rotterdam), Sara Machado (London School of Economics), Diogo Nogueira Leite (Faculty of Medicine, University of Porto), Nova Health Economics and Management Knowledge Centre, the Portuguese Association of Health Economics, the Health Economists' Study Group, and the Erasmus Choice Modelling Centre.

## Author Contributions

**Conceptualization:** Luís Filipe, Sara Valente de Almeida, Eduardo Costa, Joana Gomes da Costa, Francisca Vargas Lopes, João Vasco Santos.

**Data curation:** Luís Filipe, Sara Valente de Almeida, Eduardo Costa, Joana Gomes da Costa, Francisca Vargas Lopes, João Vasco Santos.

**Formal analysis:** Luís Filipe, Sara Valente de Almeida, Eduardo Costa, Joana Gomes da Costa, Francisca Vargas Lopes, João Vasco Santos.

**Investigation:** Luís Filipe, Sara Valente de Almeida, Eduardo Costa, Joana Gomes da Costa, Francisca Vargas Lopes, João Vasco Santos.

**Methodology:** Luís Filipe, Sara Valente de Almeida, Eduardo Costa, Joana Gomes da Costa, Francisca Vargas Lopes, João Vasco Santos.

**Project administration:** Luís Filipe.

**Supervision:** Luís Filipe.

**Validation:** Luís Filipe.

**Visualization:** Luís Filipe, João Vasco Santos.

**Writing – original draft:** Luís Filipe, Sara Valente de Almeida, Eduardo Costa, Joana Gomes da Costa, Francisca Vargas Lopes, João Vasco Santos.

**Writing – review & editing:** Luís Filipe, Sara Valente de Almeida, Eduardo Costa, Joana Gomes da Costa, Francisca Vargas Lopes, João Vasco Santos.

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
