## [Decision Letter · Decision Letter 0]

28 Jul 2022

PONE-D-22-17412Opening to deaths: A discrete choice experiment about Covid-19’s policy preferences in PortugalPLOS ONE

Dear Dr. Filipe,

Thank you for submitting your manuscript to PLOS ONE. After careful consideration, we feel that it has merit but does not fully meet PLOS ONE’s publication criteria as it currently stands. Therefore, we invite you to submit a revised version of the manuscript that addresses the points raised during the review process.

We look forward to receiving your revised manuscript.

Kind regards,

Karyn Morrissey

Academic Editor

PLOS ONE

Journal Requirements:

"We are grateful to Nova Health Economics and Management Knowledge Centre, the Portuguese

Association of Health Economics and the Erasmus Choice Modelling Centre. This report is independent research funded by the National Institute for Health Research (NIHR) Applied Research Collaboration North West Coast (ARC NWC). The views expressed in this publication are those of the authors and not necessarily those of the NHS, NIHR, or the Department of Health and Social Care. There are neither financial nor non-financial conflicts. No ethical considerations apply."

"This report is independent research funded by the National Institute for Health Research (NIHR) Applied Research Collaboration North West Coast (ARC NWC). The views expressed in this publication are those of the authors and not necessarily those of the NHS, NIHR, or the Department of Health and Social Care. There are neither financial nor non-financial conflicts. No ethical considerations apply."

5. Please ensure that you refer to Figure 1 in your text as, if accepted, production will need this reference to link the reader to the figure.

Reviewers' comments:

Reviewer's Responses to Questions

**Comments to the Author**

1. Is the manuscript technically sound, and do the data support the conclusions?

Reviewer #1: Partly

Reviewer #2: Partly

2. Has the statistical analysis been performed appropriately and rigorously? 

Reviewer #1: I Don't Know

Reviewer #2: Yes

3. Have the authors made all data underlying the findings in their manuscript fully available?

Reviewer #1: No

Reviewer #2: Yes

4. Is the manuscript presented in an intelligible fashion and written in standard English?

Reviewer #1: Yes

Reviewer #2: Yes

5. Review Comments to the Author

Reviewer #1: Comments for the Author

Thank you for the opportunity to review this manuscript. This study used DCE methods to examine the preferences pertaining to consequences of COVID-19 policies in Portugal. There have been other studies using DCEs (in other countries) that have examined consequences from COVID-19 policies, however the manuscript has referenced these studies appropriately. This study offers a different perspective from other the DCEs investigating COVID-19 policies in that the study is unique for the experience in Portugal, but more interestingly, they have conducted the study at a later timepoint where the population have already experienced prior lockdowns/COVID-19 restrictions, thereby capturing sentiment in subsequent waves. There are comments, which should be addressed by the authors, and changes that need clarification.

General comments

• The term ‘willingness to pay (WTP)’ is used to describe the marginal rate of substitution (MRS) for attribute levels, using death (as a continuous variable) as the numeraire. Using the term WTP is confusing as a cost attribute were not used in the DCE. The authors should change the wording throughout the manuscript to reflect that the MRS were measured rather than WTP.

• Data were collected when vaccination were available. Did the survey collect data about vaccination? Is this a factor that may potentially affect the results given that this study reports on the preferences of subsequent waves? This could be included in the discussion section.

Section comments

Introduction

• Line 63: “We find that avoiding deaths ……” is one of the results found. Please delete this line from the introduction.

• Lines 64-65: see comment below regarding the data analysis (under methodology) pertaining to method used to assess heterogeneity.

Methodology

• Data analysis: The major limitation in this manuscript pertains to the analysis used, which was conditional logit modelling, followed by subgroup analyses to explore heterogeneity. Although conditional logit models are typically used to initially analyse DCE data, and there are better methods that could be used when exploring heterogeneity. The authors should consider using other methods e.g., mixed logit modelling or latent class analysis, to explore heterogeneity within the sample. I’ve noted some references the authors may wish to review1-4. Including these analyses would require a substantial change to the manuscript; however, this would improve the manuscript and interpretation pertaining to heterogeneity.

• Under statistical analysis, please state the software used to analyse the data; and under survey (line 171-172) include the software used to create the D-optimal design.

• Section 3.2:

o Line 117-118: “The attributes were chosen based on international literature (Chorus 2020 and Reed 2020)”; as these are DCEs it would be better to state that other DCEs were used to inform this study.

o Line 120: Change “attributed” to “attribute”.

o Line 120-122: Is there a reference that could be included referencing the policies at the time of the survey?

• Survey, line 173: Were any changes made after the pilot study? If yes, please note the changes that were made. Was the data from the pilot used in the final analysis of the DCE?

• Table S1: move the example of the choice set into the main text rather than the appendix. Noting that the choice question is likely in Portuguese, could you translate and include the question respondents were asked to answer in their choice task? This is good for a reader to get an overview of what was explicitly asked to do in the DCE.

• Line 186-187: The sentence “We divided our analysis of the respondents characteristics into three groups: individual, household, and work-related aspects” is unclear. Can you clarify what this refers to and how these groupings were used? I’m assuming each term is an umbrella term for other more specific terms, which should be clarified in the text.

• Lines 193-195: Euros are denoted after the numeric figure and should be placed before the number.

• Did you collect any information from the respondents about their experience in completing this DCE? Was there any feedback they had given about any of the survey? If not, add as a note in the limitations section.

• Lines 215-217: The results for the conditional logit indicate that the levels for death were treated categorically. Can you please clarify this in the text? This is clarified for the MRS analysis, but not the conditional logit analysis.

• Section 3.4: the model equations available in the appendix are perhaps not needed. It may be better to reference a textbook which readers can refer to. Alternatively, specify the equations to be specific for this DCE if they wish to retain.

Results

• The main results section may need to be rewritten. The difference in the levels should be explained relative to the base level, which is not clear in the text, especially if readers are not familiar with DCEs. For example, relative to 50 deaths, respondents do not prefer 250 deaths, or something to that effect.

• line 226: Reword text “which serve as baseline” to the “which serves as the baseline level”.

• Line 226-227: “education and life restrictions have statistically indistinguishable coefficients for levels medium and high.” This is unclear, can you please clarify this text.

• Table 3: Both the conditional logit (using death as a categorical variable) and the MRS (using death as a continuous variable) are reported in this table. I suggest merging Table 3 and Table S3 in the appendix to form one table e.g., models across the column headings and attributes and levels as rows, then move this table to the appendix. For the results, graph in a forest plot, as this is better visually for a reader. Please report model statistics models for the results (e.g., number of observations, AIC, BIC, loglikelihood) in the table.

• Figure 1: Willingness-to-pay (WTP) by subgroups: It is difficult to read the bars for each group, can you redo the bars so that teach of the groups are not overlapping? Alternatively present the more interesting one in the main body (gender and workplace), and make these larger, and move other categories to the appendix.

• Line 248-249: Two characteristics are noted as having statistically significant dfferences in preference. Under analysis, can you include what tests were run to determine statistical signifcance and report the results in the text in this section.

• Line 253: as you have used a continuous variable, please rephrase the sentence such that the “lower number of deaths avoided” is quantified as per the MRS.

Discussion

• Line 262 and line 290: Change the term “utility” to “preferences” as this was the outcome being estimated. Please change all instances throughout manuscript.

• Line 302: A comparison with the general population is made with reference to income. However, the baseline characteristics table does not report the comparative income for the general population. I can understand this may be the case, but I think this should be rephrased somewhat or a reference included noting what the mean or median income is for Portugal.

• The higher risk of poverty is an interesting finding pertaining to the MRS results. The 45% level being described as a catastrophic scenario and the implications, could be described in more detail in the discussion. If space is an issue, the discussion pertaining to women could be shortened.

• Line 272-274: Rephrase sentence “During the pandemic ….”

• Another limitations to include in the discussion is that the highest level for deaths in the DCE was below the maximum level.

1 Lancsar E, Fiebig DG, Hole AR. Discrete Choice Experiments: A Guide to Model Specification, Estimation and Software. PharmacoEconomics 2017; 35: 697-716.

2 Bridges JF, Hauber AB, Marshall D, et al. Conjoint analysis applications in health--a checklist: a report of the ISPOR Good Research Practices for Conjoint Analysis Task Force. Value Health 2011; 14: 403-413.

3 Hauber AB, González JM, Groothuis-Oudshoorn CG, et al. Statistical Methods for the Analysis of Discrete Choice Experiments: A Report of the ISPOR Conjoint Analysis Good Research Practices Task Force. Value Health 2016; 19: 300-315.

4 Reed Johnson F, Lancsar E, Marshall D, et al. Constructing experimental designs for discrete-choice experiments: report of the ISPOR Conjoint Analysis Experimental Design Good Research Practices Task Force. Value Health 2013; 16: 3-13.

Reviewer #2: Comments for authors

This is an interesting paper on public preferences for COVID controls using DCEs. I have some major and some minor comments. My major concerns are around the design and implementation of the survey and the way you present some of the results.

Major comments

1. I’m not sure your title “Opening to deaths” really encapsulates the breadth of analysis in your paper.

2. In terms of the duration covered, you used “January to June 2021”, it would be good to see the choice card/full questionnaire in Portuguese to be able to see what the respondents were looking at. I have to say that this is a little bit vague to me and could be interpreted as either being 5 or 6 months depending on how you read it. Was there a reason you chose 6 months? I also find it a bit strange that you are asking about the past as well to some – as this was conducted until March 2021.

3. I have some concerns over the designs and levels. First, can you comment on having both a poverty level and income level in the same choice experiment. These are both income levels at the end of the day – I wonder if you can reflect on that (and if e.g. employment might be better to use if this is to be repeated?).

4. In terms of the educational impairment – why was a % of students used rather than a level of education days lost? The latter seems to be more of a realistic measure. Surely most are affected in some way by a lockdown.

5. In Table 2 you present the descriptive statistics for income as being “household income” but in the text this is presented as being “per individual” – this is a bit confusing. Is there not also a potential issue that this income level has been impacted by COVID - did you consider asking the “pre-COVID” level? This might affect how you interpret the results in terms of the household income loss being presented as percentages… and this is also I imagine the reason you are not able to report national level percentage rates (though you could use the latest Census?).

6. It is good to see that the authors obtained ethical clearance from an ethics board in Portugal. I just want the authors to confirm that there was approval for data collection with those under 18 (as 28.79% of the sample in Table 2 is being reported as being <18). If not explicit, then the authors may need to clarify that they have such approval from the Nova Business School ethics committee – I realise open surveys of this type are hard to control. You might want to rephrase line 347 – there clearly are ethical considerations in this kind of survey!

7. I must say I find it a bit surprising that given the age distribution of the sample that only 28% have no college degree – presumably the majority of under 18s and about half of the next age group would not be of the age to have a college degree.

8. I find it strange that you use “Willingness to Pay” and use a deaths metric as the basis for it (rather than a euro value). Maybe you could rephrase this? “Death acceptance”? (though admittedly there is a difference between “willingness to accept” and “willingness to pay” – I have a feeling here that acceptance may be a better phrase!). When you talk of things like “women have a lower WTP than men” in my mind that suggests they value life less, when in fact this is the opposite!

9. I am wondering if it would be worth thinking about what your results would imply if you translated this (via the income change) to a Value of a Prevented Fatality – a rough calculation I have done suggests the numbers you are getting are pretty low (which might reflect the perception that older and those more likely to die sooner are those most at risk).

10. Given that this was very much a time of much change in terms of COVID regulation and information, I am wondering if you have considered including examining whether these preferences had any time element – e.g. if there was any systematic difference between the responses in March compared to those in January.

11. I think there is a question you may want to think about in terms of uncertainty of a pandemic spread on health. In your experiment you use fixed numbers – when the epidemiological modelling of risked deaths would probably be much higher (given the deaths on a day when things were controlled was higher than your 250 number). It might be worth reflecting on this and the implications this has on the use of DCEs in this context. DCEs do take some time to design and implement/analyse – so compared to focus groups and other forms of surveys this may restrict their applicability. Sampling, as you note, is also important (given your sample is not representative).

Minor comments:

1. Line 27 and Line 326 – I don’t think “auscultate” is quite the right word – my understanding is that this is more a medical term (I will have to admit I had to look up what it meant). Perhaps consider using “assess” instead – which I think conveys what you mean.

2. Line 305 – “the NHS” – not clear what this is (I’m presuming you mean the SNS in Portugal but would be better to specify – as you also use the NHS on line 346 to refer to UK NHS).

6. PLOS authors have the option to publish the peer review history of their article (what does this mean?). If published, this will include your full peer review and any attached files.

Reviewer #1: No

Reviewer #2: No

---

## [Author Response · Author response to Decision Letter 0]

12 Sep 2022

September 11, 2022

“Trade-offs during the COVID-19 pandemic: a discrete choice experiment about policy preferences in Portugal”

Previously titled as:

“Opening to deaths: A discrete choice experiment about Covid-19’s policy preferences in Portugal”

Dear Editors,

We thank the editors and reviewers for providing us with valuable and constructive feedback, as well as the opportunity to submit a revised version of our manuscript.

We believe that the paper is now clearer and stronger. The main changes in the revised version include:

1. The inclusion of a latent class model analysis to account for class preference heterogeneity

2. Changes in graphs and table to enhance comprehensibility

3. Changes in terminology that improve the technical consistency of the paper

Please find below the detailed answers to the comments raised by each of the reviewers. Note that references to pages and lines are according to the clean version of the manuscript. Thank you for your consideration of our manuscript. 

Reviewer #1 

General comments

1. The term ‘willingness to pay (WTP)’ is used to describe the marginal rate of substitution (MRS) for attribute levels, using death (as a continuous variable) as the numeraire. Using the term WTP is confusing as a cost attribute were not used in the DCE. The authors should change the wording throughout the manuscript to reflect that the MRS were measured rather than WTP.

Response: Thank you for the excellent comment. We understand that using WTP without a traditional cost attribute may be confusing. As such, we follow the recommendation of the reviewer and substitute WTP by MRS across the entire text. 

2. Data were collected when vaccination were available. Did the survey collect data about vaccination? Is this a factor that may potentially affect the results given that this study reports on the preferences of subsequent waves? This could be included in the discussion section.

Response: Thank you for the comment. During the survey collection period, vaccination was at a very early stage - in the peak of the third wave, only about 2.5% of the population had received a first dose of the vaccine. For this reason, no information was collected regarding vaccination status. We do not believe that vaccination affected results substantially for two main reasons. During the study period, the coverage rate of vaccines was very low (and focused only on elderly groups of the population). Moreover, there was no clear consensus regarding the efficacy of the vaccine - which was only verified some months after the survey was completed. We included this in the discussion section (page 22, line 386).

Section comments

Introduction

4. Line 63: “We find that avoiding deaths ……” is one of the results found. Please delete this line from the introduction.

Response: Thank you for noticing, we deleted the sentence.

5. Lines 64-65: see comment below regarding the data analysis (under methodology) pertaining to method used to assess heterogeneity.

Response: Thank you for the excellent comment. Given the reviewer’s following comments we decided to add a latent class model (with 3 classes) to the analysis. We rewrote the introduction section accordingly (page 4, lines 78), and also adjusted the methodology (page 12, line 235) and discussion sections (page 20, line 341).

Methodology

6. Data analysis: The major limitation in this manuscript pertains to the analysis used, which was conditional logit modelling, followed by subgroup analyses to explore heterogeneity. Although conditional logit models are typically used to initially analyse DCE data, and there are better methods that could be used when exploring heterogeneity. The authors should consider using other methods e.g., mixed logit modelling or latent class analysis, to explore heterogeneity within the sample. I’ve noted some references the authors may wish to review1-4. Including these analyses would require a substantial change to the manuscript; however, this would improve the manuscript and interpretation pertaining to heterogeneity.

Response: Thank you for noticing and making such a relevant suggestion. The reviewer is right and a subgroup analysis is not enough to account for preference heterogeneity. We therefore decided to include an extra section with a latent class model with 3 classes (based on CAIC and BIC) (page 17, line 286). Our paper is now in the style of Chorus (2020), with a conditional logit, an unconditional subgroup analysis and a latent class model. We decided in favour of a latent class model in detriment of a mixed logit because we believe that accounting for class preferences’ heterogeneity better fits the problem we are trying to address. We also tested a latent class model with 2 classes. The results are similar to the 3-class model but without the “garbage class” (class 3). Therefore, we decided to refer only to the 3-class model in the text. 

7. Under statistical analysis, please state the software used to analyse the data; and under survey (line 171-172) include the software used to create the D-optimal design.

Response: Thank you for your comment. The statistical analysis was conducted using STATA 16 software. This information is now included in the manuscript (in the methodology section).

8. Section 3.2: Line 117-118: “The attributes were chosen based on international literature (Chorus 2020 and Reed 2020)”; as these are DCEs it would be better to state that other DCEs were used to inform this study.

Response: Thank you for noticing, the text was revised based on your suggestion (page 6, line 133).

Line 120: Change “attributed” to “attribute”.

Response: Thank you for your comment. This typo was corrected in the manuscript.

o Line 120-122: Is there a reference that could be included referencing the policies at the time of the survey?

Response: Thank you for your comment. We have included a reference in the manuscript to describe the policies implemented by the Portuguese government during the pandemic. 

9. Survey, line 173: Were any changes made after the pilot study? If yes, please note the changes that were made. Was the data from the pilot used in the final analysis of the DCE?

Response: No, no changes were made after the pilot. The pilot was used to inform the priors of the D-Optimal design and was included in the analysis. We added a sentence (page 9, line 196) clarifying that we use the pilot as part of the overall dataset in analysis, 

10. Table S1: move the example of the choice set into the main text rather than the appendix. Noting that the choice question is likely in Portuguese, could you translate and include the question respondents were asked to answer in their choice task? This is good for a reader to get an overview of what was explicitly asked to do in the DCE.

Response: Thank you for the excellent comment. The choice set was moved to the main text - in the methodology section. This choice set is translated to English from its original version (in Portuguese). The original version (in Portuguese) was also added to the appendix. Figures S1 and S2 now display the instructions to the DCE and the one choice set with the opt-out possibility. 

11. Line 186-187: The sentence “We divided our analysis of the respondents characteristics into three groups: individual, household, and work-related aspects” is unclear. Can you clarify what this refers to and how these groupings were used? I’m assuming each term is an umbrella term for other more specific terms, which should be clarified in the text.

Response: Thank you for your comment. The text was revised and that particular sentence was deleted.

12. Lines 193-195: Euros are denoted after the numeric figure and should be placed before the number.

Response: Thank you for your suggestion. The text was revised accordingly.

13. Did you collect any information from the respondents about their experience in completing this DCE? Was there any feedback they had given about any of the survey? If not, add as a note in the limitations section.

Response: Thank you for your comment. We did not collect feedback regarding the survey completion. This limitation was included in the manuscript (page 23, line 405). 

14. Lines 215-217: The results for the conditional logit indicate that the levels for death were treated categorically. Can you please clarify this in the text? This is clarified for the MRS analysis, but not the conditional logit analysis.

Response: Thank you for your comment. The description of the analysis was revised, and included in the methodology section. It now explicitly mentions that the conditional logit model, used for the first analysis, was estimated with all attribute levels coded as binary (page 12, line 236).

15. Section 3.4: the model equations available in the appendix are perhaps not needed. It may be better to reference a textbook which readers can refer to. Alternatively, specify the equations to be specific for this DCE if they wish to retain.

Response: Thank you for your comment. The model equations were removed from the appendix, and references were included in the main text. 

Results

16. The main results section may need to be rewritten. The difference in the levels should be explained relative to the base level, which is not clear in the text, especially if readers are not familiar with DCEs. For example, relative to 50 deaths, respondents do not prefer 250 deaths, or something to that effect.

Response: Thank you for your comment. We agree that not including the reference levels in the text could be misleading. The entire section has been altered to always refer to the baseline levels for each attribute.

17. line 226: Reword text “which serve as baseline” to the “which serves as the baseline level”.

Response: Thank you for the suggestion, which we have implemented. 

18. Line 226-227: “education and life restrictions have statistically indistinguishable coefficients for levels medium and high.” This is unclear, can you please clarify this text.

Response: Thank you for your comment. The text was revised to clarify this issue. 

19. Table 3: Both the conditional logit (using death as a categorical variable) and the MRS (using death as a continuous variable) are reported in this table. I suggest merging Table 3 and Table S3 in the appendix to form one table e.g., models across the column headings and attributes and levels as rows, then move this table to the appendix. For the results, graph in a forest plot, as this is better visually for a reader. Please report model statistics models for the results (e.g., number of observations, AIC, BIC, loglikelihood) in the table.

Response: Thank you for your relevant comment. As suggested, we have moved table 3 to the appendix. Now the table has three columns, showing: first, the coefficients of the model that considers deaths as a categorical variable; second, the coefficients of the model with deaths as a continuous variable; third, the MRS computed from the latter, using deaths as numeraire. In the main text we now present the forest plots for these three columns.

20. Figure 1: Willingness-to-pay (WTP) by subgroups: It is difficult to read the bars for each group, can you redo the bars so that teach of the groups are not overlapping? Alternatively present the more interesting one in the main body (gender and workplace), and make these larger, and move other categories to the appendix.

Response: Thank you, we have updated the figures according to the reviewer’s suggestions. 

21. Line 248-249: Two characteristics are noted as having statistically significant differences in preference. Under analysis, can you include what tests were run to determine statistical significance and report the results in the text in this section.

Response: Thank you for your comments. The conclusions were taken visually by looking at the 95% confidence intervals. In order to fit the page, we do not include the confidence intervals in table S3, in appendix. But we believe that with the changes the reviewer suggested they should now be easier to assess in the graphs.

22. Line 253: as you have used a continuous variable, please rephrase the sentence such that the “lower number of deaths avoided” is quantified as per the MRS.

Response: Thank you for your comment. We have rewritten the sentence, to clarify it.

Discussion

23. Line 262 and line 290: Change the term “utility” to “preferences” as this was the outcome being estimated. Please change all instances throughout manuscript.

Response: Thank you for your comment. We have revised the manuscript accordingly. 

24. Line 302: A comparison with the general population is made with reference to income. However, the baseline characteristics table does not report the comparative income for the general population. I can understand this may be the case, but I think this should be rephrased somewhat or a reference included noting what the mean or median income is for Portugal.

Response: Thank you for your comment. While we have evidence of the high proportion of high-educated individuals, we are only hypothesizing its link to income. Therefore, we have rewritten the sentence, accordingly.

25. The higher risk of poverty is an interesting finding pertaining to the MRS results. The 45% level being described as a catastrophic scenario and the implications, could be described in more detail in the discussion. If space is an issue, the discussion pertaining to women could be shortened.

Response: Thank you for your comment. We added the discussion about the poverty attribute in the discussion, as suggested (page 19, line 315)

26. Line 272-274: Rephrase sentence “During the pandemic ….”

Response: Thank you for your comment. We have rewritten the sentence.

27. Another limitation to include in the discussion is that the highest level for deaths in the DCE was below the maximum level.

Response: Thank you for your comment. In fact, the COVID-19 pandemic was characterized by substantial uncertainty. The DCE setting implies using fixed levels for each attribute, throughout the experiment. These levels are defined based on the literature and on existing expectations. Some of these levels may become obsolete at a given moment, considering ongoing changes in the pandemic situation. This limitation was now included in the discussion section (page 22, line 379). 

Reviewer #2: Comments for authors

Major comments

28. I’m not sure your title “Opening to deaths” really encapsulates the breadth of analysis in your paper.

Response: Thank you for your comment. We adjusted the title of the paper to be more precise. 

29. In terms of the duration covered, you used “January to June 2021”, it would be good to see the choice card/full questionnaire in Portuguese to be able to see what the respondents were looking at. I have to say that this is a little bit vague to me and could be interpreted as either being 5 or 6 months depending on how you read it. Was there a reason you chose 6 months? I also find it a bit strange that you are asking about the past as well to some – as this was conducted until March 2021.

Response: Thank you for your comment. The instructions given to the respondents and a choice set (in Portuguese) were added in appendix (Figure S1 and S2). We make it explicit in the instruction that we mean 6 months. The reason to choose 6 months came from our perception of what would be a realistic timeframe for the Covid-19 associated policies, while minimising potential recall bias. In 2020, the restrictions imposed by the governments started to ease during the summer. We assumed they would follow the same trend in 2021. 

The reviewer concern with the survey being distributed during the hypothetical period of the experience is a valid one. While a DCE is hypothetical (and we make that clear in the instructions), the respondents could have been influenced by the reality they were living when answering the survey. For example, the reviewer pointed out, in another comment, that the levels of daily deaths surpassed our maximum level at a certain point during the period of the survey. We have extended the discussion to include these topics.

30. I have some concerns over the designs and levels. First, can you comment on having both a poverty level and income level in the same choice experiment. These are both income levels at the end of the day – I wonder if you can reflect on that (and if e.g. employment might be better to use if this is to be repeated?).

Response: Thank you for your comment. When designing the study, we decided to include both attributes (poverty level and income level) since they are expected to capture two different dimensions - which are not necessarily related. While the income level refers to the respondent’s individual income, the poverty level refers to the average society poverty level. While the income attribute captures a more “selfish” perspective, the poverty attribute captures a more “altruistic” dimension.

31. In terms of the educational impairment – why was a % of students used rather than a level of education days lost? The latter seems to be more of a realistic measure. Surely most are affected in some way by a lockdown.

Response: Thank you for your excellent comment. We agree that the number of education days lost is a more realistic measure. However, when designing the study, we were afraid that the average respondent would have difficulties in understanding the concept. For this reason, we opted for using the % of students impaired, which should be seen as a proxy for the number of education days lost. We believe that this variable is easier to grasp by the respondents.

32. In Table 2 you present the descriptive statistics for income as being “household income” but in the text this is presented as being “per individual” – this is a bit confusing. Is there not also a potential issue that this income level has been impacted by COVID - did you consider asking the “pre-COVID” level? This might affect how you interpret the results in terms of the household income loss being presented as percentages… and this is also I imagine the reason you are not able to report national level percentage rates (though you could use the latest Census?).

Response: Thank you for your comment. This is a very interesting point. However, to avoid an overly long survey, we restricted the number of questions asked to the most relevant characteristics. For this reason, no information regarding the pre-Covid income levels was collected. The definition of household income was not clearly explained in the first version of the manuscript. In the survey, we collected information on each respondent’s household income, as well as on household characteristics. We then estimated, based on the size and composition of the household, the equivalised income of each individual. We have rewritten the descriptive statistics accordingly to make this distinction clearer. 

33. It is good to see that the authors obtained ethical clearance from an ethics board in Portugal. I just want the authors to confirm that there was approval for data collection with those under 18 (as 28.79% of the sample in Table 2 is being reported as being <18). If not explicit, then the authors may need to clarify that they have such approval from the Nova Business School ethics committee – I realise open surveys of this type are hard to control. You might want to rephrase line 347 – there clearly are ethical considerations in this kind of survey!

Response: Thank you for your comment. In fact, open surveys such as this one are able to collect responses from potentially any individual. The previous version of the manuscript had a typo: the sample included very few answers from individuals under 18. Nonetheless, we eliminated all answers from individuals below 18 years old. These records were destroyed from the original database. The new analysis has been conducted only for individuals with at least 18 years old, and the results have been revised accordingly - although no substantial changes are identified. 

34. I must say I find it a bit surprising that given the age distribution of the sample that only 28% have no college degree – presumably the majority of under 18s and about half of the next age group would not be of the age to have a college degree.

Response: Thank you for noticing. As mentioned before, there was a typo in the manuscript that we regret. We have corrected this and the new version includes a revised table and description.

35. I find it strange that you use “Willingness to Pay” and use a deaths metric as the basis for it (rather than a euro value). Maybe you could rephrase this? “Death acceptance”? (though admittedly there is a difference between “willingness to accept” and “willingness to pay” – I have a feeling here that acceptance may be a better phrase!). When you talk of things like “women have a lower WTP than men” in my mind that suggests they value life less, when in fact this is the opposite!

Response: Thank you for the comment. The term ‘willingness to pay (WTP)’ was used to describe the marginal rate of substitution (MRS) for attribute levels, using death as the numeraire. We understand that using WTP, without a traditional cost attribute may be misleading. As such, we follow the recommendation of both reviewers and substitute WTP by MRS across the entire text - which is a more precise measure.

36. I am wondering if it would be worth thinking about what your results would imply if you translated this (via the income change) to a Value of a Prevented Fatality – a rough calculation I have done suggests the numbers you are getting are pretty low (which might reflect the perception that older and those more likely to die sooner are those most at risk).

Response: Thank you for your question. We agree this is a very valid point and something we also thought about. However, the story of the paper is about to what extent the preferences of the respondents were in line with the policies adopted during the Covid-19 pandemic. While we agree that measuring the statistical value of life would be interesting, we also think it would require a longer discussion and a longer set of references. Also, it would require certainty of generalizability. For space concerns and objectivity of our message we decided not to include a discussion about the statistical value of life.

Still, as the reviewer points out, (a very rough) back of the envelope calculation would yield approximately 1.8 million euros per life, which would be close to the lower intervals of the values generally found for developed countries. We believe this could have been caused by a “normalisation” of deaths during the period in analysis, as many may have seen some deaths as “inevitable” during the pandemic. Also, as the reviewer points out, the average age of the people who were dying was high, which may have impacted these numbers.

37. Given that this was very much a time of much change in terms of COVID regulation and information, I am wondering if you have considered including examining whether these preferences had any time element – e.g. if there was any systematic difference between the responses in March compared to those in January.

Response: Thank you for your comment. This is an excellent point - and it was aligned with our initial idea (to analyse how responses changes over time, and whether we could identify any pattern). However, given the data collection mechanism, such analysis was not possible to perform. As with many online surveys, we had a peak of answers in the first weeks after launching the survey, followed by a very low number of answers in later weeks. For this reason, we do not have a significant number of observations to perform a statistical analysis with a time dimension.

38. I think there is a question you may want to think about in terms of uncertainty of a pandemic spread on health. In your experiment you use fixed numbers – when the epidemiological modelling of risked deaths would probably be much higher (given the deaths on a day when things were controlled was higher than your 250 number). It might be worth reflecting on this and the implications this has on the use of DCEs in this context. DCEs do take some time to design and implement/analyse – so compared to focus groups and other forms of surveys this may restrict their applicability. Sampling, as you note, is also important (given your sample is not representative).

Response: Thank you for your comment. You raise very relevant and important points. We have extended the discussion section to include these topics (page22, line 379). 

Minor comments:

39. Line 27 and Line 326 – I don’t think “auscultate” is quite the right word – my understanding is that this is more a medical term (I will have to admit I had to look up what it meant). Perhaps consider using “assess” instead – which I think conveys what you mean.

Response: Thank you. We agree with the suggestion, the text was revised accordingly.

40. Line 305 – “the NHS” – not clear what this is (I’m presuming you mean the SNS in Portugal but would be better to specify – as you also use the NHS on line 346 to refer to UK NHS).

Response: Thank you for the comment. The text was revised accordingly, we now refer to the “Portuguese NHS” instead of to the “NHS”.

---

## [Decision Letter · Decision Letter 1]

1 Nov 2022

PONE-D-22-17412R1Trade-offs during the COVID-19 pandemic: a discrete choice experiment about policy preferences in PortugalPLOS ONE

Dear Dr. Filipe,

Thank you for submitting your manuscript to PLOS ONE. After careful consideration, we feel that it has merit but does not fully meet PLOS ONE’s publication criteria as it currently stands. Therefore, we invite you to submit a revised version of the manuscript that addresses the points raised during the review process.

We look forward to receiving your revised manuscript.

Kind regards,

Karyn Morrissey

Academic Editor

PLOS ONE

Journal Requirements:

Reviewers' comments:

Reviewer's Responses to Questions

**Comments to the Author**

1. If the authors have adequately addressed your comments raised in a previous round of review and you feel that this manuscript is now acceptable for publication, you may indicate that here to bypass the “Comments to the Author” section, enter your conflict of interest statement in the “Confidential to Editor” section, and submit your "Accept" recommendation.

Reviewer #1: (No Response)

2. Is the manuscript technically sound, and do the data support the conclusions?

Reviewer #1: Yes

3. Has the statistical analysis been performed appropriately and rigorously? 

Reviewer #1: Yes

4. Have the authors made all data underlying the findings in their manuscript fully available?

Reviewer #1: Yes

5. Is the manuscript presented in an intelligible fashion and written in standard English?

Reviewer #1: No

6. Review Comments to the Author

Reviewer #1: Comments for the Author

Thank you for making changes to your initial manuscript. The change in title is better than the original title proposed. The authors have addressed some of the main issues, but I think the paper needs another revision before it is ready for publication.

The main points I made in the initial review pertain to the analyses that were conducted. Although the authors didn’t include a mixed logit analysis, they have revised the analysis to include a latent class analysis which improves the manuscript. The conditional logit (CL) analysis and corresponding subgroup analyses extending from the CL models have been retained in the manuscript but are not sufficient to explain the inferences made concerning heterogeneity, which the authors do note in their response.

There are other issues I’ve noted with respect to the MRS results (which may be typos) that need to be reconciled. For example, the abstract notes “Estimates suggest that individuals would be willing to sacrifice 20% of their income to avoid 47 deaths per day….”; however, results in the table denote this should probably be for the level of household income being 30% (Table S2). The authors should check all the results in the tables correspond with the text. There are other instances in the abstract and the results (on page 14 paragraph 1 lines 261-267) where this is inconsistent and should be updated.

On page 17 (lines 281-282, and 284-285) the wording with respect to the comparison between the subgroups such as men and women is difficult to understand - “This means that women, …… and everyday life restrictions”. Giving an interpretation for both groups within the subgroup, similar to what you have when you are discussing the main results e.g., (assuming 2 corresponds to the mid-level and 3 corresponds to the worst level perceived) would improve readability e.g., the women were willing to sacrifice 20% of their income to avoid seven daily deaths, whereas the men were willing to sacrifice 20% of their income to avoid 17 daily deaths.

There are still quite a few typos and grammar issues, for example, consistency in the tense used and numeric formatting, and I have noted a few of these below.

Other comments

Line 95-96: “during three fortnights” is odd phrasing; perhaps change to “over a six week period”.

Where numeric figures are below 10, please write out value in full e.g., line 104 – “5” to “five”.

The numeric formatting should be consistent throughout the manuscript e.g., line 99 “1.516”, line 215 “1,250 euro”, line 219 “€1750”, lines 220-224 where decimal places in the text pertaining to percentages are written as either no decimal places or to one or two decimal places; remove the decimal from all numeric figures that are in the thousands throughout. e.g., “1.516”, “6.994”. etc.

Line 141-142: “ the attribute of percentage of impaired students is used as a proxy for “number of education days lost…”. This is not a major issue, but maybe add something about the framing of this attribute, as I don’t think it does refer to the number of education days lost. The basis of the attribute appears to be ‘person-based and does not quite reflect “hours lost”. I’d probably add this as a limitation in the discussion if you want to discuss it further.

Overall phrasing under Section 3.4 p10: I think some of the wording used in the manuscript should be changed to make it easier to read. For example:

Line 211: “Respondents’ age ranged from 18 up to over 85 years old” to “Respondents aged 18 years and over”;

Line 211: Delete the words “Concentration is high in young ages”

Line 214-215: “Considering on the differences in household’s size and composition, individuals display a median monthly equivalised income of around 1,250 euros”. It should be made clear whether it is the study sample or the population.

Results and attributes and levels: “social restrictions” and “life restrictions” are used interchangeably. Are these the same? What was given to the respondents? That probably should be used consistently in the manuscript.

Line 283: “observed” would be better to use than “verified”. Another instance in the discussion.

Latent class analysis:

In your response I can see you tested models with 2 or 3 classes. Did you test whether model fit using CAIC and BIC beyond three latent classes? You should add a statement in your manuscript confirming whether the three class model was the best model fit based on CAIC and BIC parameters and the range of model with different classes tested.

The summary of results requires more careful review. For example in line 292, it states that Class 1 is “more affected by household income losses and poverty losses”; which is not quite the case. A stronger aversion to household income reducing by 30% is observed in Class 3 (reference class) compared to Class 1.

In line 293-294, the text noting “It (Class 1) has a higher prevalence of males… “ should be reworded. For example “people with Class 1 preferences a more likely to be male and work remotely”. In lines 343-345 in the discussion i.e., “higher share of males …..”. Please rephrase as noted above. People with Class X preferences are more likely to have characteristics such as ……

Is there a reference you used for calculating the relative importance of the attributes as shown in Figure 3?

7. PLOS authors have the option to publish the peer review history of their article (what does this mean?). If published, this will include your full peer review and any attached files.

Reviewer #1: No

---

## [Author Response · Author response to Decision Letter 1]

16 Nov 2022

Reviewer #1 

Thank you for making changes to your initial manuscript. The change in title is better than the original title proposed. The authors have addressed some of the main issues, but I think the paper needs another revision before it is ready for publication.

We thank the reviewer for the comment! The reviewers’ contributions have been fundamental to create a better version of this manuscript.

The main points I made in the initial review pertain to the analyses that were conducted. Although the authors didn’t include a mixed logit analysis, they have revised the analysis to include a latent class analysis which improves the manuscript. The conditional logit (CL) analysis and corresponding subgroup analyses extending from the CL models have been retained in the manuscript but are not sufficient to explain the inferences made concerning heterogeneity, which the authors do note in their response.

We thank the reviewer for the comment! We included a latent class model because we believed that class heterogeneity would better accommodate our research question. The rational has to do with the way society has split into groups of people, in which for some, avoiding deaths was prioritized while others advocated those negative economic impacts should be kept to a minimum. We follow the same methodology as Chorus 2020, published in this journal.

There are other issues I’ve noted with respect to the MRS results (which may be typos) that need to be reconciled. For example, the abstract notes “Estimates suggest that individuals would be willing to sacrifice 20% of their income to avoid 47 deaths per day….”; however, results in the table denote this should probably be for the level of household income being 30% (Table S2). The authors should check all the results in the tables correspond with the text. There are other instances in the abstract and the results (on page 14 paragraph 1 lines 261-267) where this is inconsistent and should be updated.

We thank the reviewer for the comment! There are no typos, but we agree that the sentences are not clear. Therefore, we made clarifications both in the abstract and the results’ section. For example, the sentence in the abstract now reads as: “Estimates suggest that individuals would be willing to sacrifice 30% instead of 10% of their income to avoid approximately 47 deaths per day during the first 6 months of 2021” Note that the text is now also clear in stating that the numbers are approximations (to avoid decimal cases).

On page 17 (lines 281-282, and 284-285) the wording with respect to the comparison between the subgroups such as men and women is difficult to understand - “This means that women, …… and everyday life restrictions”. Giving an interpretation for both groups within the subgroup, similar to what you have when you are discussing the main results e.g., (assuming 2 corresponds to the mid-level and 3 corresponds to the worst level perceived) would improve readability e.g., the women were willing to sacrifice 20% of their income to avoid seven daily deaths, whereas the men were willing to sacrifice 20% of their income to avoid 17 daily deaths.

We thank the reviewer for the recommendation! We have revised the text accordingly. 

There are still quite a few typos and grammar issues, for example, consistency in the tense used and numeric formatting, and I have noted a few of these below.

Other comments

Line 95-96: “during three fortnights” is odd phrasing; perhaps change to “over a six week period”.

We thank the reviewer for the correction! The text was revised accordingly.

Where numeric figures are below 10, please write out value in full e.g., line 104 – “5” to “five”.

We thank the reviewer for the correction! The text was revised accordingly. 

The numeric formatting should be consistent throughout the manuscript e.g., line 99 “1.516”, line 215 “1,250 euro”, line 219 “€1750”, lines 220-224 where decimal places in the text pertaining to percentages are written as either no decimal places or to one or two decimal places; remove the decimal from all numeric figures that are in the thousands throughout. e.g., “1.516”, “6.994”. etc.

We thank the reviewer for identifying those text inconsistencies! The text was revised accordingly. 

Line 141-142: “ the attribute of percentage of impaired students is used as a proxy for “number of education days lost…”. This is not a major issue, but maybe add something about the framing of this attribute, as I don’t think it does refer to the number of education days lost. The basis of the attribute appears to be ‘person-based and does not quite reflect “hours lost”. I’d probably add this as a limitation in the discussion if you want to discuss it further.

We thank the reviewer for the comment! We agree that saying this attribute is a proxy for days lost may come as a little bit of a stretch. Therefore, we deleted the sentence in Line 141-142. We have rephrased it to “To some extent, this percentage reflects the inequalities associated with schools’ closure, distance learning and other mitigation strategies.”, which is the correct reason for this attribute being included. This sentence is now in Page 8, Line 161-162. 

Overall phrasing under Section 3.4 p10: I think some of the wording used in the manuscript should be changed to make it easier to read. For example: Line 211: “Respondents’ age ranged from 18 up to over 85 years old” to “Respondents aged 18 years and over”;

We thank the reviewer for the comment! The text has been revised accordingly.

Line 211: Delete the words “Concentration is high in young ages”

We thank the reviewer for the suggestion! The text has been revised accordingly.

Line 214-215: “Considering on the differences in household’s size and composition, individuals display a median monthly equivalised income of around 1,250 euros”. It should be made clear whether it is the study sample or the population.

We thank the reviewer for the suggestion! A clarification has been made.

Results and attributes and levels: “social restrictions” and “life restrictions” are used interchangeably. Are these the same? What was given to the respondents? That probably should be used consistently in the manuscript.

We thank the reviewer for the correction! The reviewer is correct. In the Portuguese survey the question asked about life restrictions. To make the text consistent with the question asked to the respondents, we have updated the text to contain only the term “life restrictions”.

Line 283: “observed” would be better to use than “verified”. Another instance in the discussion.

We thank the reviewer for the suggestion The text has been revised accordingly.

Latent class analysis:

In your response I can see you tested models with 2 or 3 classes. Did you test whether model fit using CAIC and BIC beyond three latent classes? You should add a statement in your manuscript confirming whether the three class model was the best model fit based on CAIC and BIC parameters and the range of model with different classes tested.

We thank the reviewer for the comment! We tested the model from 2 up to 10 latent classes. A clarification if made in the analysis section (Page 13, line 241)

The summary of results requires more careful review. For example in line 292, it states that Class 1 is “more affected by household income losses and poverty losses”; which is not quite the case. A stronger aversion to household income reducing by 30% is observed in Class 3 (reference class) compared to Class 1.

We thank the reviewer for the comment. Indeed, that sentence is not correct and is therefore misleading. The entire section was revised to be more clear about whether the inference is relative to another class or is of absolute value.

In line 293-294, the text noting “It (Class 1) has a higher prevalence of males… “ should be reworded. For example “people with Class 1 preferences a more likely to be male and work remotely”. 

We thank the reviewer for the suggestion! The entire section has been revised to accommodate for the type of wording suggested by the reviewer.

In lines 343-345 in the discussion i.e., “higher share of males …..”. Please rephrase as noted above. People with Class X preferences are more likely to have characteristics such as ……

We thank the reviewer for the suggestion! The text has been revised accordingly.

Is there a reference you used for calculating the relative importance of the attributes as shown in Figure 3?

We thank the reviewer for the question! Figure 3 values are computed by taking the absolute value of the maximum level of each attribute and dividing them by the sum of the absolute value of all the maximum levels of each attribute, per class. This type of graph is common in other DCE papers, since it is very simple way to illustrate relative preferences. One example would be Arslan et al. 2020, that we added in our references.

Regarding figure 3, however, we noted there was a computing mistake regarding the denominator of class 3. The mistake has been corrected and the figure and respective descriptions are now updated. We also made some minor quality of life improvements that should make the figure easier to interpret.

---

## [Editor Report · Decision Letter 2]

18 Nov 2022

Trade-offs during the COVID-19 pandemic: a discrete choice experiment about policy preferences in Portugal

PONE-D-22-17412R2

Dear Dr. Filipe,

We’re pleased to inform you that your manuscript has been judged scientifically suitable for publication and will be formally accepted for publication once it meets all outstanding technical requirements.

Kind regards,

Karyn Morrissey

Academic Editor

PLOS ONE
---

## [Editor Report · Acceptance letter]

9 Dec 2022

PONE-D-22-17412R2 

Trade-offs during the COVID-19 pandemic: a discrete choice experiment about policy preferences in Portugal 

Dear Dr. Filipe:

I'm pleased to inform you that your manuscript has been deemed suitable for publication in PLOS ONE. Congratulations! Your manuscript is now with our production department. 

Kind regards, 

on behalf of

Dr. Karyn Morrissey 

Academic Editor

PLOS ONE